# GATING OUT SENSORY NOISE IN A SPIKE-BASED LONG SHORT-TERM MEMORY NETWORK

## ABSTRACT

Spiking neural networks are being investigated both as biologically plausible models of neural computation and also as a potentially more efficient type of neural network. While convolutional spiking neural networks have been demonstrated to achieve near state-of-the-art performance, only one solution has been proposed to convert gated recurrent neural networks, so far. Recurrent neural networks in the form of networks of gating memory cells have been central in state-of-the-art solutions in problem domains that involve sequence recognition or generation. Here, we design an analog gated LSTM cell where its neurons can be substituted for efficient stochastic spiking neurons. These adaptive spiking neurons implement an adaptive form of sigma-delta coding to convert internally computed analog activation values to spike-trains. For such neurons, we approximate the effective activation function, which resembles a sigmoid. We show how analog neurons with such activation functions can be used to create an analog LSTM cell; networks of these cells can then be trained with standard backpropagation. We train these LSTM networks on a noisy and noiseless version of the original sequence prediction task from Hochreiter & Schmidhuber (1997), and also on a noisy and noiseless version of a classical working memory reinforcement learning task, the T-Maze. Substituting the analog neurons for corresponding adaptive spiking neurons, we then show that almost all resulting spiking neural network equivalents correctly compute the original tasks.

## 1 INTRODUCTION

With the manifold success of biologically inspired deep neural networks, networks of spiking neurons are being investigated as potential models for computational and energy efficiency. Spiking neural networks mimic the pulse-based communication in biological neurons, where in brains, neurons spike only sparingly – on average 1-5 spikes per second (Attwell & Laughlin, 2001). A number of successful convolutional neural networks based on spiking neurons have been reported (Esser et al., 2016; Neil et al., 2016; Diehl et al., 2015; O'Connor et al., 2013; Hunsberger & Eliasmith, 2015), with varying degrees of biological plausibility and efficiency. Still, while spiking neural networks have thus been applied successfully to solve image-recognition tasks, many deep learning algorithms use recurrent neural networks (RNNs), in particular using Long Short-Term Memory (LSTM) layers (Hochreiter & Schmidhuber, 1997). Compared to convolutional neural networks, LSTMs use memory cells to store selected information and various gates to direct the flow of information in and out of the memory cells. To date, the only spike-based version of LSTM has been realized for the IBM TrueNorth platform Shrestha et al.: this work proposes a method to approximate LSTM specifically for the constrains of this neurosynaptic platform by means of a store-and-release mechanism that synchronizes the modules. This translates to a frame-based rate coding computation, which is less biological plausible and energy efficient than an asynchronous approach, as the one proposed here.

Here, we demonstrate a gated recurrent spiking neural network that corresponds to an LSTM unit with a memory cell and an input gate. Analogous to recent work on spiking neural networks (O'Connor et al., 2013; Diehl et al., 2015; Zambrano & Bohte, 2016; Zambrano et al., 2017), we first train a network with modified LSTM units that computes with analog values, and show how this LSTM-network can be converted to a spiking neural network using adaptive stochastic spiking neurons that encode and decode information in spikes using a form of sigma-delta coding (Yoon, 2016; Zambrano & Bohte, 2016; O'Connor & Welling, 2016). In particular, we develop a binary version of the adaptive

sigma-delta coding proposed in (Zambrano & Bohte, 2016): we approximate the shape of the transfer function that this model of fast-adapting spiking neurons exhibits, and we assemble the analog LSTM units using just this transfer function. Since input-gating is essential for maintaining memorized information without interference from unrelated sensory inputs (Hochreiter & Schmidhuber, 1997), and to reduce complexity, we model a limited LSTM neuron consisting of an input cell, input gating cell, a Constant Error Carousel (CEC) and output cell. The resultant analog LSTM network is then trained on a number of classical sequential tasks, such as the noise-free and noisy Sequence Prediction and the T-Maze task (Hochreiter & Schmidhuber, 1997; Bakker, 2002). We demonstrate how nearly all the corresponding spiking LSTM neural networks correctly compute the same function as the analog version.

Note that the conversion of gated RNNs to spike-based computation implies a conversion of the neural network from a time step based behavior to the continuous-time domain: for RNNs, this means having to consider the continuous signal integration in the memory cell. We solve the time conversion problem by approximating analytically the spiking memory cell behavior through time.

Together, this work is a first step towards using spiking neural networks in such diverse and challenging tasks like speech recognition and working memory cognitive tasks.

## 2 MODEL

To construct an Adapting Spiking LSTM network, we first describe the Adaptive Spiking Neurons and we approximate the corresponding activation function. Subsequently, we show how an LSTM network comprised of a spiking memory cell and a spike-driven input-gate can be constructed and we discuss how analog versions of this LSTM network are trained and converted to spiking versions.

**Adaptive Spiking Neuron.** The spiking neurons that are used in this paper are Adaptive Spiking Neurons (ASNs) as described in Bohte (2012). This is a variant of an adapting Leaky Integrate & Fire (LIF) neuron model that includes fast adaptation to the dynamic range of input signals. The ASNs used here communicate with spikes of a fixed height $h = 1$ (binary output), as suggested by Zambrano et al. (2017). The behavior of the ASN is determined by the following equations:

incoming postsynaptic current:
$$I(t) = \sum_i \sum_{t_s^i} w_i \exp\left(\frac{t_s^i - t}{\tau_\beta}\right); \tag{1}$$

input signal:
$$S(t) = (\phi * I)(t); \tag{2}$$

threshold:
$$\vartheta(t) = \vartheta_0 + \sum_{t_s} m_f \vartheta(t_s) \exp\left(\frac{t_s - t}{\tau_\gamma}\right); \tag{3}$$

internal state:
$$\hat{S}(t) = \sum_{t_s} \vartheta(t_s) \exp\left(\frac{t_s - t}{\tau_\eta}\right), \tag{4}$$

where $w_i$ is the weight (synaptic strength) of the neuron's incoming connection; $t_s^i < t$ denote the spike times of neuron $i$, and $t_s < t$ denote the spike times of the neuron itself; $\phi(t)$ is an exponential smoothing filter with a short time constant $\tau_\phi$; $\vartheta_0$ is the resting threshold; $m_f$ is a variable controlling the speed of spike-rate adaptation; $\tau_\beta, \tau_\gamma, \tau_\eta$ are the time constants that determine the rate of decay of $I(t), \vartheta(t)$ and $\hat{S}(t)$ respectively (see Bohte (2012) and Zambrano & Bohte (2016) for more details). As in Bohte (2012), the ASN emits spikes following a stochastic firing condition defined as:

$$\lambda(V(t), \vartheta(t)) = \lambda_0 \exp\left(\frac{V(t) - \vartheta(t)/2}{\Delta V}\right), \tag{5}$$

where $V(t)$ is the membrane potential defined as the difference between $S(t)$ and $\hat{S}(t)$, $\lambda_0 = 0.005$ is a normalization parameter and $\Delta V = 0.1$ is a scaling factor that defines the slope of the stochastic area.

**Activation function of the Adaptive Analog Neuron.** In order to create a network of ASNs that performs correctly on typical LSTM tasks, our approach is to train a network of Adaptive Analog

Neurons (AANs) and then convert the resulting analog network into a spiking one, similar to O'Connor et al. (2013); Diehl et al. (2015); Zambrano & Bohte (2016). We define the activation function of the AANs as the function that maps the input signal $S$ to the average PSC $I$ that is perceived by the *next* (receiving) ASN in a defined time window. We normalize the obtained spiking activation function at the point where it reaches a plateau. We then fit the normalized spiking activation function with a sum-of-exponentials shaped function as:

$$\text{AAN}(S) = \frac{1}{a \cdot \exp(b \cdot S) + c \cdot \exp(d \cdot S) + 1}, \tag{6}$$

with derivative:

$$\frac{\text{dAAN}(S)}{\text{d}S} = -\frac{a \cdot b \cdot \exp(b \cdot S) + c \cdot d \cdot \exp(d \cdot S)}{(a \cdot \exp(b \cdot S) + c \cdot \exp(d \cdot S) + 1)^2}, \tag{7}$$

where, for the neuron parameters used, we find $a = 148.7$, $b = -10.16$, $c = 3.256$ and $d = -1.08$.

Using this mapping from the AAN to the ASN (see Figure 1), the activation function can be used during training: thereafter, the ASNs are used as "drop in" replacements for the AANs in the trained network. Unless otherwise stated, the ASNs use $\tau_\eta = \tau_\beta = \tau_\gamma = 10$ ms, and $\vartheta_0$ and $m_f$ are set to 0.3 and 0.18 for all neurons. The spike height, $h$, is found such that $\text{ASN}(4.8) = 1$. Note that the spike height $h$ is a normalization parameter for the activation function of the ASN model: in order to have binary communication across the network, the output weights are simply scaled by $h$.

**Adaptive Spiking LSTM.** An LSTM cell usually consists of an input and output gate, an input and output cell and a CEC Hochreiter & Schmidhuber (1997). Deviating from the original formulation, and more recent versions where forget gates and peepholes were added Gers et al. (2002), the Adaptive Spiking LSTM as we present it here only consists of an input gate, input and output cells, and a CEC. As noted, to obtain a working Adaptive Spiking LSTM, we first train its analog equivalent, the Adaptive Analog LSTM. Figure 2 shows the schematic of the Adaptive Analog LSTM and its spiking analogue. It is important to note that we aim for a one-on-one mapping from the Adaptive Analog LSTM to the Adaptive Spiking LSTM. This means that while we train the Adaptive Analog LSTM network with the standard time step representation, the conversion to the continuous-time spiking domain is achieved by presenting each input for a time window of size $\Delta t$.

**Sigmoidal ASN.** The original formulation of LSTM uses sigmoidal activation functions in the input gate and input cell. However, the typical activation function of real neurons resembles a half-sigmoid and we find that the absence of a gradient for negative input is problematic during training. Here, we approximate a sigmoidal-shaped activation function by exploiting the stochastic firing condition of the ASN. Indeed, Figure 1 shows that the ASN has a non-null probability to fire even under the threshold $\vartheta_0$. Therefore, the AAN transfer function of Eq. 6 holds a gradient in that area. Together

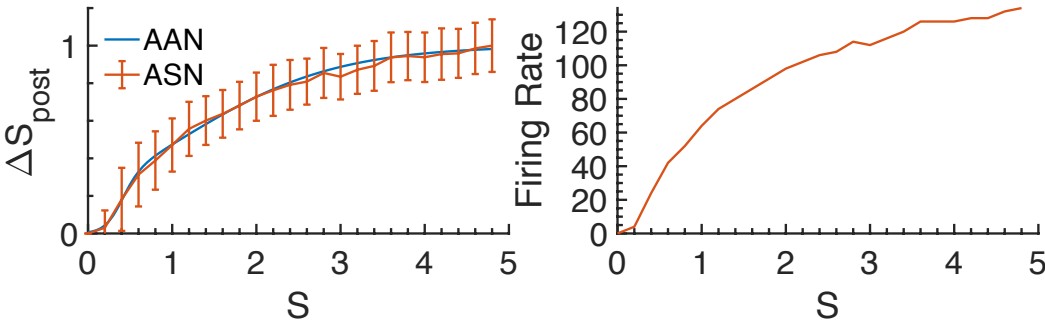

Figure 1: The average output signal of the stochastic ASN (left) as a function of its incoming PSC $I$, where the error bars indicate the standard deviation of the spiking simulation, and the corresponding AAN curve. The shape of the ASN curve is well described by the AAN activation function, Equation 6. The ASN firing rate (Hz) for different values of input signal (right).

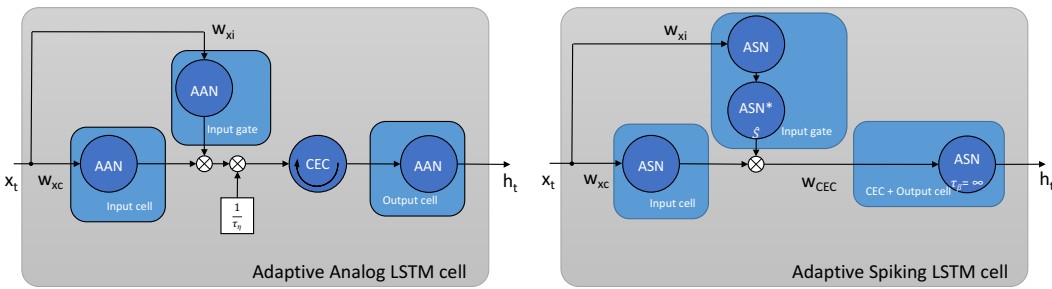

Figure 2: An overview of the construction of an Adaptive Analog LSTM (left) and an Adaptive Spiking LSTM cell. This compares to an LSTM with only an input gate.

with the maximal activation being normalized to 1 (see Eq. 6 for $\lim_{S \to \infty}$) the AAN transfer function represents a good candidate for LSTM operations such as closing and opening the gates.

**Spiking input gate and spiking input cell.**    The AAN functions are used in the Adaptive Analog LSTM cell for the input gate and input cell. The activation value of the input cell is multiplied by the activation value of the input gate, before it enters the CEC, see Figure 2. In the spiking version of the input gate, the outgoing signal from the ASN is accumulated in an intermediate neuron (ASN* in Figure 2). The internal state $\hat{S}$ of this neuron is then multiplied with the spikes that move from the ASN of the input cell to the ASN of the output cell. This leads to a direct mapping from the Adaptive Analog LSTM to the Adaptive Spiking LSTM.

**Spiking Constant Error Carousel (CEC) and spiking output cell.**    The Constant Error Carousel (CEC) is the central part of the LSTM cell and avoids the vanishing gradient problem Hochreiter & Schmidhuber (1997). In the Adaptive Spiking LSTM, we merge the CEC and the output cell to one ASN with an internal state that does not decay – in the brain could be implemented by slowly decaying (seconds) neurons (Denève & Machens, 2016). The value of the CEC in the Adaptive Analog LSTM corresponds with state $I$ of the ASN output cell in the Adaptive Spiking LSTM.

In the Adaptive Spiking LSTM, we set $\tau_\beta$ in Equation 1 to a very large value for the CEC cell to obtain the integrating behavior of a CEC. Since no forget gate is implemented this results in a spiking CEC neuron that fully integrates its input. When $\tau_\beta$ is set to $\infty$, every incoming spike is added to a non-decaying PSC $I$. So if the state of the sending neuron (ASN$_{in}$ in Figure 3) has a stable inter-spike interval (ISI), then $I$ of the receiving neuron (ASN$_{out}$) is increased with incoming spike height $h$ every ISI, so $\frac{h}{ISI}$ per time step. For a stochastic neuron, this corresponds to the average increase per time step.

The same integrating behavior needs to be translated to the analog CEC. Since the CEC cell of the Adaptive Spiking LSTM integrates its input $S$ every time step by $\frac{S}{\tau_\eta}$, we can map this to the CEC of the Adaptive Analog LSTM. The CEC of a traditional LSTM without a forget gate is updated every time step by $\text{CEC}(t) = \text{CEC}(t-1) + S$, with $S$ its input value. The CEC of the Adaptive Analog LSTM is updated every time step by $\text{CEC}(t) = \text{CEC}(t-1) + \frac{S}{\tau_\eta}$. This is depicted in Figure 2 via a weight after the input gate with value $\frac{1}{\tau_\eta}$. To allow a correct continuous-time representation after the spike-coding conversion, we divide the incoming connection weight to the CEC, $W_{CEC}$, by the time window $\Delta t$. In our approach then, we train the Adaptive Analog LSTM as for the traditional LSTM (without the $\tau_\eta$ factor), which effectively corresponds to set a continuous-time time window $\Delta t = \tau_\eta$. Thus, to select a different $\Delta t$, in the spiking version $W_{CEC}$ has to be set to $W_{CEC} = \tau_\eta / \Delta t$.

The middle plot in Figure 3 shows that setting $\tau_\beta$ to $\infty$ for ASN$_{out}$ in a spiking network results in the same behavior as using an analog CEC that integrates with $\text{CEC}(t) = \text{CEC}(t-1) + S$, since the slope of the analog CEC is indeed the same as the slope of the spiking CEC. Here, every time step in the analog experiment corresponds to $\Delta t = 200$ ms. However, the spiking CEC still produces an error with respect to the analog CEC (the error increases for lower $\Delta t$s, e.g. it doubles when going from 200ms to 50ms). This is because of two reasons: first, the stochastic firing condition results in an irregular ISI; second, the adapting behavior of the ASN produces a transitory response that is not

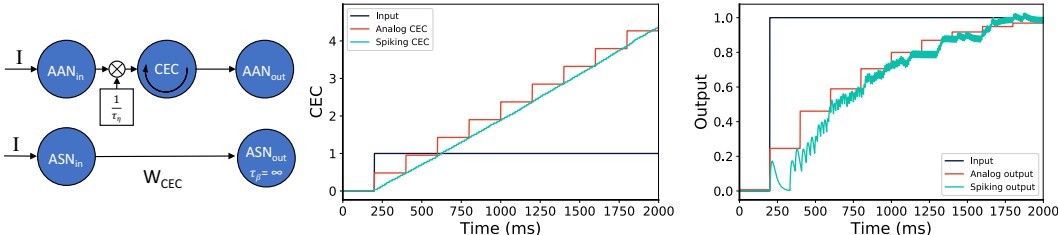

Figure 3: A simulation to illustrate how the analog CEC integrates its input signal with the same speed as an ASN with $\tau_\beta = \infty$ provided that the input signal does not change and that 1 analog time step corresponds to $\Delta t = 200$ms (middle). In the right panel, the spiking output signal approximates the analog output, in particular for high values of current.

represented by the AAN transfer function. For these reasons, by choosing bigger time windows $\Delta t$ more stable responses are obtained.

**Learning rule used for training the spiking LSTM**   To train the analog LSTMs on the supervised tasks, a customized truncated version of real-time recurrent learning (RTRL) was used. This is the same algorithm used in Gers et al. (2002), where the partial derivatives w.r.t. the weights $W_{xc}$ and $W_{xi}$ (see Figure 2) are truncated. For the reinforcement learning (RL) tasks we used RL-LSTM Bakker (2002), which uses the same customized, truncated version of RTRL that was used for the supervised tasks. RL-LSTM also incorporates eligibility traces to improve training and Advantage Learning Harmon & Baird III (1996). All regular neurons in the network are trained with traditional backpropagation.

## 3   EXPERIMENTS

Since the presented Adaptive Analog LSTM only has an input gate and no output or forget gate, we present four classical tasks from the LSTM literature that do not rely on these additional gates.

**Sequence Prediction with Long Time Lags.**   The main concept of LSTM, the ability of a CEC to maintain information over long stretches of time, was demonstrated in Hochreiter & Schmidhuber (1997) in a Sequence Prediction task: the network has to predict the next input of a sequence of $p + 1$ possible input symbols denoted as $a_1, ..., a_{p-1}, a_p = x, a_{p+1} = y$. In the *noise free* version of this task, every symbol is represented by the $p + 1$ input units with the $i - th$ unit set to 1 and all the others to 0. At every time step a new input of the sequence is presented. As in the original formulation, we train the network with two possible sequences, $(x, a_1, a_2, ..., a_{p-1}, x)$ and $(y, a_1, a_2, ..., a_{p-1}, y)$, chosen with equal probability. For both sequences the network has to store a representation of the first element in the memory cell for the entire length of the sequence ($p$). We train 20 networks on this task for a total of $100k$ trials, with $p = 100$, on an architecture with $p + 1$ input units and $p + 1$ output units. The input units are fully connected to the output units without a hidden layer. The same sequential network construction method from the original paper was used to prevent the "abuse problem": the Adaptive Analog LSTM cell is only included in the network after the error stops decreasing Hochreiter & Schmidhuber (1997). In the *noisy* version of the sequence prediction task, the network still has to predict the next input of the sequence, but the symbols from $a_1$ to $a_{p-1}$ are presented in random order and the same symbol can occur multiple times. Therefore, only the final symbols $a_p$ and $a_{p+1}$ can be correctly predicted. This version of the sequence prediction task avoids the possibility that the network learns local regularities in the input stream. We train 20 networks with the same architecture and parameters of the previous task, but for $200k$ trials. For both noise-free and noisy tasks we considered the network converged when the average error over the last 100 trials was less than $0.25$.

**T-Maze task.**   In order to demonstrate the generality of our approach, we trained a network with Adaptive Analog LSTM cells on a Reinforcement Learning task, originally introduced in Bakker (2002). In the T-Maze task, an agent has to move inside a maze to reach a target position in order to

Table 1: Summary of the results. For the Sequence Prediction tasks the number of iterations corresponds to the number of episodes (shown for the original Hochreiter & Schmidhuber (1997) and current implementation); while for the T-Maze tasks it corresponds to the total number of steps Bakker (2002). ASN accuracy (%), total number of spikes per task and firing rate (Hz) are also reported. Note that the firing rate for both the sequence prediction tasks are computed without taking into account the input and output neurons not active in a specific time frame.

| Task | Orig. Conv. It. (%) | AAN Conv. It. (%) | ASN (%) | $N_{spikes}$ (Hz) |
|---|---|---|---|---|
| Seq. Prediction | 5040 (100) | 2574 (95) | 95 | 7129(71) |
| noisy Seq. Prediction | 5680 (100) | 8732 (95) | 95 | 7114(71) |
| T-maze $N = 20$ | $1M$ (100) | 8990(100) | 100 | 2844(25) |
| noisy T-Maze $N = 20$ | $1.75M$ (100) | 71531(80) | 80 | 4261(38) |

be rewarded while maintaining information during the trial. The maze is composed of a long corridor with a T-junction at the end, where the agent has to make a choice based on information presented at the start of the task. The agent receives a reward of $4$ if it reaches the target position and $-0.4$ if it moves against the wall. If it moves to the wrong direction at the T-junction it also receives a reward of $-0.4$ and the system is reset. The larger negative reward value, w.r.t. the one used in Bakker (2002), is chosen to encourage Q-values to differentiate more during the trial. The agent has 3 inputs and 4 outputs corresponding to the 4 possible directions it can move to. At the beginning of the task the input can be either $011$ or $110$ (which indicates on which side of the T-junction the reward is placed). Here, we chose the corridor length $N = 20$. A noiseless and a noisy version of the task were defined: in the noiseless version the corridor is represented as $101$, and at the T-junction $010$; in a noisy version the input in the corridor is represented as $a0b$ where $a$ and $b$ are two uniformly distributed random variables in a range of $[0, 1]$. While the noiseless version can be learned by LSTM-like networks without input gating Rombouts et al. (2012), the noisy version requires the use of such gates. The network consists of a fully connected hidden layer with 12 AAN units and 3 Adaptive Analog LSTMs. To increase the influence of the LSTM cell in the network, we normalized the activation functions of the AAN output cell and ASN output cell at $S = 1$. The same training parameters are used as in Bakker (2002); we train 20 networks for each task and all networks have the same architecture. As a convergence criteria we checked whenever the network reached on average a total reward greater than $3.5$ in the last 100 trials.

## 4 RESULTS

As shown in Table 1, all of the networks that were successfully trained for the noise-free and noisy Sequence Prediction tasks could be converted into spiking networks. Figure 4 shows the last 6 inputs of a noise-free Sequence Prediction task before (left) and after (right) the conversion, demonstrating the correct predictions made in both cases. Indeed, for the 19 successful networks, after presenting either $x$ or $y$ as the first symbol of the sequence, the average error over the last 200ms was always below the chosen threshold of $0.25$. As it can be seen in Figure 6, the analog and the spiking CEC follow a comparable trend during the task, reaching similar values at the end of the simulation. Note that, in the noisy task, all the successfully trained networks were still working after the conversion: in this case, due to the input noise, the CEC values are always well separated. Finally, we found that the number of trials needed to reach the convergence criterion were, on average, lower than the one reported in Hochreiter & Schmidhuber (1997).

Similar results were obtained for the T-Maze task: all the networks were successful after the conversion in both the noise-free and noisy conditions. Figure 5 shows the Q-values of a noisy T-Maze task, demonstrating the correspondence between the analog and spiking representation even in presence of noisy inputs. However, we notice that the CEC of the spiking LSTMs reach different values compared to their analog counterparts. This is probably due to the increased network and task complexity.

In general, we see that the spiking CEC value is close to the analog CEC value, while always exhibiting some deviations. Moreover, Table 1 reports the average firing rate computed per task, showing reasonably low values compatible with the one recorder from real neurons.

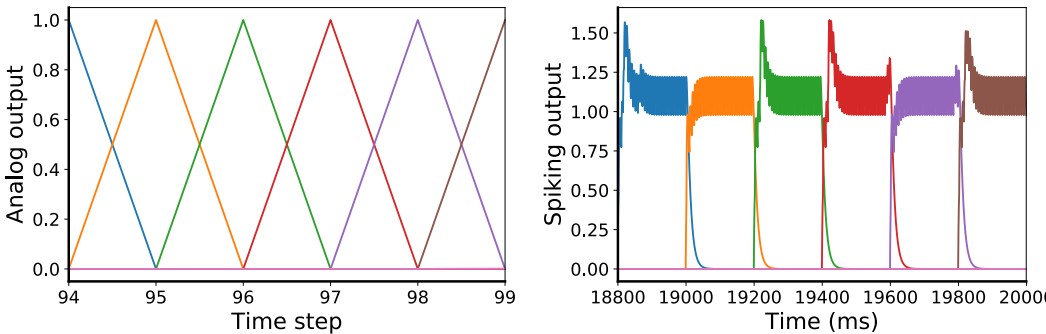

Figure 4: The output values of the analog (left) and spiking (right) network for the noise-free Sequence Prediction task. Only the last 6 input symbols of the series are shown. The last symbol $y$ (brown) is correctly predicted either in the last time step (analog) and in the last 200ms (spiking).

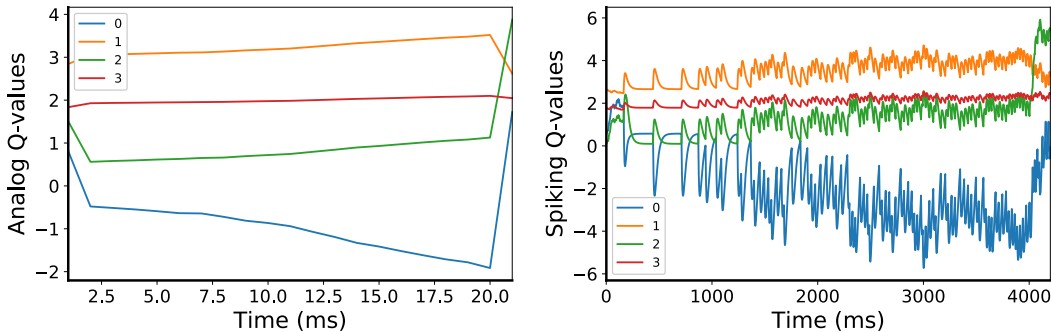

Figure 5: The Q-values of the analog (left) and spiking (right) network for the noisy T-Maze task. In the last 200ms it correctly selects the right action (blue line).

## 5 DISCUSSION

Gating is a crucial ingredient in recurrent neural networks that are able to learn long-range dependencies Hochreiter & Schmidhuber (1997); Cho et al. (2014). Input gates in particular allow memory cells to maintain information over long stretches of time regardless of the presented - irrelevant - sensory input Hochreiter & Schmidhuber (1997). The ability to recognize and maintain information for later use is also that which makes gated RNNs like LSTM so successful in the great many sequence related problems, ranging from natural language processing to learning cognitive tasks Bakker (2002).

To transfer deep neural networks to networks of spiking neurons, a highly effective method has been to map the transfer function of spiking neurons to analog counterparts and then, once the network has been trained, substitute the analog neurons with spiking neurons O'Connor et al. (2013); Diehl et al. (2015); Zambrano & Bohte (2016). Here, we showed how this approach can be extended to gated memory units, and we demonstrated this for an LSTM network comprised of an input gate and a CEC. Hence, we effectively obtained a low-firing rate asynchronous LSTM network.

The most complex aspect of a gating mechanism turned out to be the requirement of a differentiable gating function, for which analog networks use sigmoidal units. We approximated the activation function for a stochastic Adaptive Spiking Neurons, which, as many real neurons, approximates a half-sigmoid (Fig. 1). We showed how the stochastic spiking neuron has an effective activation even below the resting threshold $\vartheta_0$. This provides a gradient for training even in that area. The resultant LSTM network was then shown to be suitable for learning sequence prediction tasks, both in a noise-free and noisy setting, and a standard working memory reinforcement learning task. The learned network could then successfully be mapped to its spiking neural network equivalent for at least 90% of the trained analog networks.

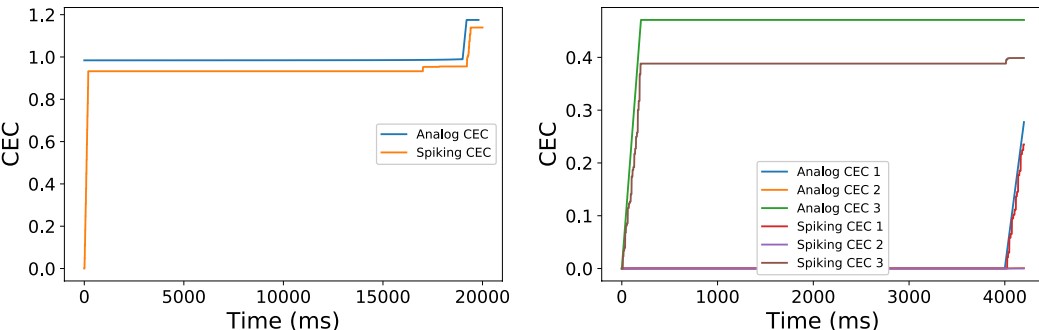

Figure 6: The values of the analog CECs and spiking CECs for the noise-free Sequence Prediction (left, only one CEC cell was used) and noise-free T-maze (right, three CEC cells were used) tasks. The spiking CEC is the internal state $\hat{S}$ of the output cell of the Adaptive Spiking LSTM.

We also showed that some difficulties arise in the conversion of analog to spiking LSTM. Principally, the ASN activation function is derived for steady-state adapted spiking neurons, and this difference causes an error that may be large for fast changing signals. Analog-valued spikes as explored in Zambrano & Bohte (2016) could likely resolve this issue, at the expense of some loss of representational efficiency.

Although the adaptive spiking LSTM implemented in this paper does not have output gates Hochreiter & Schmidhuber (1997), they can be included by following the same approach used for the input gates: a modulation of the synaptic strength. The reasons for our approach are multiple: first of all, most of the tasks do not really require output gates; moreover, modulating each output synapse independently is less intuitive and biologically plausible than for the input gates. A similar argument can be made for the forget gates, which were not included in the original LSTM formulation: here, the solution consists in modulating the decaying factor of the CEC.

Finally, which gates are really needed in an LSTM network is still an open question, with answers depending on the kind of task to be solved (Greff et al., 2017; Zaremba, 2015). For example, the AuGMEnT framework does not use gates to solve many working memory RL tasks (Rombouts et al., 2012). In addition, it has been shown by Cho et al. (2014); Chung et al. (2014); Greff et al. (2017) that a combination of input and forget gates can outperform LSTM on a variety of tasks while reducing the LSTM complexity.

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
