# OpenReview forum: "Gating out sensory noise in a spike-based Long Short-Term Memory network"
_ICLR.cc/2018/Conference — Reject_

### Official Review · AnonReviewer2 · 2017-11-27
**An interesting idea, but it seems that the main claims of are not sufficiently well proven**

**Rating:** 5
**Confidence:** 4

**Review:**

First the authors suggest an adaptive analog neuron (AAN) model which can be trained by back-propagation and then mapped to an Adaptive Spiking Neuron (ASN). Second, the authors suggest a network module called Adaptive Analog LSTM Cell (AA-LSTM) which contains input cells, input gates, constant error carousels (CEC) and output cells. Jointly with the AA-LSTM, the authors describe a spiking model (AS-LSTM) that is meant to reproduce its transfer function. It is shown quantitatively that the transfer functions of isolated AAN and AA-LSTM units are well approximated by their spiking counterparts. Two sets of experiments are reported, a sequence prediction task taken from the original LSTM paper and a T-maze task solved with reward based learning.

In general, the paper presents an interesting idea. However, it seems that the main claims of the introduction are not sufficiently well proven later. Also, I believe that the tasks are rather simple and therefore it is not demonstrated that the approach performs well on practically relevant tasks.

On general level, it should be clarified whether the model is meant to reproduce features of biology or whether the model is meant to be efficient. If the model is meant to reproduce biology, some features of the model are problematic. In particular, that the CEC is modeled with an infinitely long integration time constant of the input current. This would produce infinitely long EPSPs. However, I think there is a chance that minor changes of the model could still work while being more realistic. For example, I would find it more convincing to put the CEC into the adaptation time constants by using a large tau_gamma or tau_eta.

If the model is meant to provide efficient spiking neural networks, I find the tasks too simple and too artificial. This is particularly true in comparison to the speech recognition tasks VAD and TIMIT which were already solved in Esser et al. with spiking and efficient feedforward networks.

The authors say in the introduction that they target to model recurrent neural networks. This is an important open question. The usage of the CEC is an interesting idea toward this goal.
However, beside the presence of CEC I do not see any recurrence in the used networks. This seems in contradiction with what is implicitly claimed in the introduction, title and abstract. There are only input-output neuron connections in the sequence prediction task, and a single hidden layer for the T-maze (which does not seem to be recurrently connected). This is problematic as the authors mention that their goal is to reproduce the functionality of LSTMs with spiking neurons for which the network recurrence is an important feature.


Regarding more low-level comments:

- The authors used a truncated version of RTRL to train LSTMs and standard back-propagation for single neurons. I wonder why two different algorithms were used, as, in principle, they compute the same gradient
either forward or backward.
Is there a reason for this? Did the truncated RTRL bring any
additional benefit compared to the exact backpropagation already
implemented in automatic differentiation software?

- The sigma-delta neuron model seems quite ad-hoc and incompatible
with most simulators and dedicated hardware. I wonder whether the
AS-LSTM model would still be valid if the ASN model is replaced with a
standard SRM model for instance.

- The authors claim in the introduction that they made an analytical conversion from discrete to continuous time. I did not find this in the main text.

- The axes in Figure 1 are not defined (what is Delta S?) and the
caption does not match. "Average output signal [...] as a function of its incoming PSC I" output signal is not defined, and S is presented in the graph, but not I.

---

### Official Review · AnonReviewer3 · 2017-11-27
**A spiking implementation of LSTMs**

**Rating:** 5
**Confidence:** 3

**Review:**

The authors propose a first implementation of spiking LSTMs. This is an interesting and open problem. However, the present work somewhat incomplete, and requires further experiments and clarifications.

Pros:
1. To my best knowledge, this is the first mapping of LSTMs to spiking networks
2. The authors tackle an interesting and challenging problem.

Cons:
1. In the abstract the authors mention that another approach has been taken, but is never stated what’s the problem that this new one is trying to address. Also, H&S 1997 tested several tasks, which is the one that the authors are referring to?
2. Figure 1 is not very easy to read. The authors can spell out the labels of the axis (e.g. S could be input, S)
3. Why are output and forget gates not considered here?
4. A major point in mapping LSTMs to spiking networks is its biological plausibility. However, the authors do not seem to explore this. Of particular interest is its relationship to a recent proposal of a cortical implementation of LSTMs (Cortical microcircuits as gated-RNNs, NIPS 2017).
5. The text should be improved, for example in the abstract: “that almost all resulting spiking neural network equivalents correctly..”, please rephrase.
6. Current LSTMs are applied in much more challenging problems than the original ones. It would be important to test one of this, perhaps the relatively simple pixel-by-pixel MNIST task. If this is not feasible, please comment.

Minor comments:
1. Change in the abstract “can be substituted for” > “can be substituted by”
2. A new body of research aims at using backprop in spiking RNNs (e.g. Friedemann and Ganguli 2017). The present work gets around this by training the analog version instead. It would be of interesting to discuss how to train spiking-LSTMs as this is an important topic for future research.
3. As the main promise of using spiking nets (instead of rate) is their potential efficiency in neuromorphic systems, it would be interesting to contrast in the text the two options for LSTMs, and give some more quantitative analyses on the gain of spiking-LSTM versus rate-LSTMs in terms of efficiency.

---

### Official Review · AnonReviewer1 · 2017-12-05
**A method for converting a trained analog LSTM-type network to a spiking version.**

**Rating:** 4
**Confidence:** 4

**Review:**

Here the authors propose a variant of an analog LSTM and then further propose a mechanism by which to convert it to a spiking network, in what a computational neuroscientist would call a 'mean-field' approach.  The result is a network that communicates using only spikes.  In general I think that the problem of training or even creating spiking networks from analog networks is interesting and worthy of attention from the ML community.  However, this manuscript feels very early and I believe needs further focus and work before it will have impact in the community.

I can see three directions in which this work could be improved to provide wider interest:
1. Neurophysiological realism -  It appears the authors are not interested in this direction given the focus of the manuscript ( other than mentioning the brain as motivation).

2.  ML interest - From a pure ML point of view some interesting questions relate to training / computations / representations / performance.  However, in the manuscript the tasks trained are exceedingly simple and unconvincing from either a representations or performance perspective.  Since the main novelty of the manuscript is the 'spikification' algorithm, little is learned about how spiking networks function, or how spiking networks might represent data or implement computations.

3. Hardware considerations - There is no analysis of what has been made more efficient, more sped-up, how to meaningfully implement the algorithm, etc., etc.  A focus in this direction could find an applied audience.

As a minor comment, the paper could stand to be improved in terms of exposition.  In particular, the paper relies on ideas from other papers and the assumption is largely made that the reader is familiar with them, although the paper is self-contained.

---

### Decision · Program_Chairs · 2018-01-29
**ICLR 2018 Conference Acceptance Decision**

**Decision:**

Reject

**Comment:**

The reviewers agreed that the paper was somewhat preliminary in terms of the exposition and empirical work.  They all find the underlying problem quite interesting and challenging (i.e. spiking recurrent networks).  However, the manuscript failed to motivate the approach.  In particular, everyone agrees that spiking networks are very interesting, but it's unclear what problem the presented work is solving.  The authors need to be more clear about their motivation and then close the loop with empirical validation that their approach is solving the motivating problem (i.e. do we learn something about biological plausibility, are spiking networks better than traditional LSTMs at modeling a particular kind of data, or are they more efficiently implemented on hardware?).  Motivating the work with one of these followed by convincing experiments would make this a much stronger paper.

Pros:
- Tackles an interesting and challenging problem at the intersection of neuroscience and ML
- A novel method for creating a spiking LSTM

Cons:
- The motivation is not entirely clear
- The empirical analysis is too simple and does not demonstrate the advantages of this approach
- The paper seems unfocused and could use rewriting